# An Analysis of the Temporomandibular Joint Range of Motion and Related Factors in Children and Adolescents

**DOI:** 10.3390/children8060515

**Published:** 2021-06-17

**Authors:** YounJung Park, Taeyang Lee, Minkyeong Seog, Seong-Oh Kim, Joohee Kim, Jeong-Seung Kwon, Chung-Min Kang

**Affiliations:** 1Department of Orofacial Pain and Oral Medicine, Yonsei University College of Dentistry, Seoul 03722, Korea; darkstar@yuhs.ac (Y.P.); jskwon@yuhs.ac (J.-S.K.); 2Department of Pediatric Dentistry, Yonsei University College of Dentistry, Seoul 03722, Korea; 2sun@yuhs.ac (T.L.); janeseog@yuhs.ac (M.S.); ksodds@yuhs.ac (S.-O.K.); badybady123@yuhs.ac (J.K.)

**Keywords:** range of motion, maximum mouth opening, child, adolescent

## Abstract

This study was designed to establish safe guidelines for pediatric dental practice regarding temporomandibular joint (TMJ) range of motion (ROM) and mouth area (MA). A total of 438 children aged 3–15 years old of homogenous ethnicity participated in the study; the distribution of participants was approximately equal (sex; *n* = 15; age, *n* = 30). Maximum mouth opening (MMO), body height, weight, and age of each participant were recorded, and the TMJ ROM including anterior and lateral movements, MA, and mouth width were documented. Males showed higher mouth width, MMO, and MA values than females. MMO and MA increased with age, height, and weight in a statistically significant manner. MMO of 40 mm is reached by the age of 5.2 years, at a height of 105.9 cm and a weight of 18.6 kg. MMO showed a moderate correlation with age, height, weight, and mouth width, and MA moderately correlated with mouth width. Anterior and lateral movements did not show any close relation to these aforementioned factors. The findings of this study suggest that forcible mouth opening over 40 mm should be more cautiously considered, especially in children shorter than 105 cm, lighter than 18 kg and in children under 5 years old.

## 1. Introduction

The measurement of the temporomandibular joint (TMJ) range of motion (ROM) is a simple yet important method in the functional evaluation of the masticatory system [1]; maximum mouth opening (MMO) is a significant diagnostic reference. In clinical practice, however, mouth area (MA) measurements can be more useful than MMO measurements since retraction during dental procedures to visualize the oral cavity relates more to the planar MA than to the linear MMO.

Mouth props are routinely used when treating pediatric patients at dental clinics to enhance the quality of dental care and for patient safety. Ito et al. reported that forcible MMO using a mouth prop narrowed the upper airway diameter, which could lead to dyspnea and trigger asphyxia [2]. Therefore, safe guidelines should be established regarding pediatric ROM when using mouth props in dental practice.

ROM measurements have been reported to show a correlation with various factors such as ethnicity, sex, age, height, and weight. Chen et al. showed that the MMO increased with age, height, weight, and mouth width (MW) in a group of 518 Taiwanese children (age range, 3–5 years) [3], and Muller et al. fabricated age-related percentiles for the MMO of children based on a retrospective data sample of 20,719 Zurich children (age range, 4–17 years) [4]. However, most previous studies have analyzed only the relationship between MMO and these aforementioned factors, and have rarely considered anterior and lateral movements, and MA. Some studies included anterior and lateral movements, though these were measured in a group of participants distributed unevenly according to sex and age such as 1011 German children (age range, 10–17 years) [5] and 303 Brazilian children (age range 6–14 years) [6].

This study aimed to establish safe guidelines for TMJ ROM and MA in an evenly distributed sample size targeting all age ranges in childhood according to sex and age, and to analyze the correlation with diverse independent variables including sex, age, height, weight, body mass index (BMI), and MW in children and adolescents aged 3–15 years.

## 2. Materials and Methods

### 2.1. Participants

The study participants comprised Korean children and adolescents aged 3 to 15 years who had visited the Yonsei University Dental Hospital between July 2019 and June 2020. Inclusion criteria comprised participants with fully erupted sound primary or permanent upper and lower incisors, who were able to comprehend and perform mandibular movements as instructed, and those in general good health. Exclusion criteria comprised participants with a history of facial trauma, any pain or restriction in terms of mandibular movement, open bite or a crossbite, a prominent facial asymmetry, precocious puberty, and any systemic disease such as juvenile rheumatoid arthritis. 

In total, 440 participants were measured and the results concerning 438 participants were evaluated (females, *n* = 223; males, *n* = 215). Two participants were excluded due to precocious puberty (*n* = 1) and withdrawal of consent (*n* = 1). The distribution of the participants was designed to be as equal as possible, with 15 participants classified according to sex and 30 participants classified according to age (Figure 1). The study was approved by the Yonsei University Dental Hospital Institutional Review Board (IRB no. 2-2019-0019; approval date: 27 June 2019). The assent of a child participant and parental or guardian permission were obtained from all participants included in this study.

### 2.2. Measurement: Height, Weights and BMI

Standing height and body weight were recorded without wearing shoes and heavy garments using an anthropometric scale (DS-102, JENIX^®^, Seoul, Korea) with a precision of 0.1 cm for height and 0.1 kg for weight. The BMI was calculated based on the measured height and weight.

### 2.3. Measurement: MW

Before measuring the MW, the participants were seated in an upright position in a dental chair. A trained examiner measured the distance between oral commissures (in millimeters) three times, with the mouth in a closed resting position, using a metallic ruler to determine each participant’s MW (Figure 2a).

### 2.4. Measurement: TMJROM

The measurements of TMJ ROM including MMO, protrusion (P), right laterotrusion (RL), and left laterotrusion (LL) were performed and repeated three times by the same trained examiner, using a metallic ruler. In this study, the MMO was defined as the maximal interincisal distance on unassisted active mouth opening. MMO was obtained by verbally encouraging participants to open their mouths as far as possible. The linear interincisal measurement (in millimeters) included the distance between the fully erupted primary or permanent incisors (Figure 2b). Positive overbite was not measured in this study, and participants with a negative overbite, that is, an anterior open bite, were excluded. P, RL, and LL were recorded in millimeters following the Diagnostic Criteria for Temporomandibular Disorders (DC/TMD) Clinical Examination Protocol [7] considering horizontal overlap (overjet) for protrusion (Figure 2c) and midline deviation for laterotrusion. If a participant was uncooperative when measuring the ROM, no measurements were performed.

### 2.5. Measurement: MA

A premeasured 10-mm scale was attached to each participant’s chin as a reference. Each participant was instructed to open the mouth as wide as possible, and a clinical photo was taken to capture the open mouth. MA measurements were taken at a horizontal using a Canon 600D camera in the same room with background indicating a distance of 50 cm. If a participant was uncooperative when measuring the MA, no measurements were performed.

After transferring the photos to the computer, the outline of the inner lip on the photo was delineated (Figure 2d), and measurements of the area of the mouth entrance (in square millimeters) were obtained using a 10-mm scale reference using ImageJ version 1.52a (National Institutes of Health, Bethesda, MD, USA) software.

### 2.6. Statistical Analyses

The internal consistency among the three repeated values of MW, MMO, P, RL, LL, and MA was assessed using Cronbach’s alpha [8]. The variables of age, height, weight, BMI, and MW were grouped into quartiles. Homogeneity of variances was assessed using Levene’s test. Welch and Brown-Forsythe procedures both showed similar results in terms of analysis of variance (ANOVA) for these unequal sample sizes. Therefore, ANOVA was used to compare the MMO, P, RL, LL, and MA values within each quartile group according to age, height, weight, BMI, and MW. The sample was divided into quartile age groups with a three-year variation (3–6, 6–9, 9–12, and 12–15 years), quartile height groups (95.0–118.2, 118.2–135.4, 135.4–156.6, and 156.6–185.0 cm), quartile weight groups (13.6–21.1, 21.1–33.9, 33.9–49.2, and 49.2–110.0 kg), quartile BMI groups (10.2–15.8, 15.8–17.5, 17.5–20.7, and 20.7–34.2 kg/m^2^), and quartile MW groups (31.0–39.1, 39.1–43.0, 43.0–46.7, and 46.7–63.7 mm). Post hoc tests were performed using the least significant difference (LSD) test for pairwise comparisons. Differences in MMO, P, RL, LL, and MA values in groups classified according to sex were compared using two sample *t*-tests. MMO, P, RL, LL, and MA values were correlated with age, height, weight, BMI, and average MW using the Pearson’s correlation test. The correlation levels were assessed using a rule of thumb for interpreting the size of a correlation coefficient [9]. Multiple regression analysis was then performed. Statistical significance was set at 0.05. All statistical analyses were performed using SPSS version 25.0 (IBM Corp., Armonk, NY, USA).

## 3. Results

### 3.1. Internal Consistency

High Cronbach’s alpha values (>0.9) from three repeated MW, MMO, P, RL, LL, and MA measurements provided high reliability; hence, the average values obtained from these three readings were applied to further statistical analyses. 

### 3.2. MMO and MA According to Sex, Age, Height, Weight, BMI, and MW

The distribution of the participants was almost equal (according to sex, *n* = 15; according to age, *n* = 30). The average MMO values for males versus females aged 3–15 years were 45.9 ± 7.6 mm and 43.8 ± 6.0 mm, respectively, (*p* < 0.01). Males showed higher MW, MMO, and MA values than females (*p* < 0.05) (Table 1). 

ANOVA test results showed that the MMO and MA values differed significantly in each quartile group according to age, height, weight, BMI, and MW (*p* < 0.05) (Figure 3). Subsequently, an LSD post hoc test showed that the difference in most of the pairwise comparisons was significant within the quartile groups (*p* < 0.05). Furthermore, the logarithmic trend line suggested that the MMO increased with age, height, and weight. It also demonstrated that 40 mm is reached by the age of 5.2 years, at a standing height of 105.9 cm and bodyweight of 18.6 kg (Figure 4).

### 3.3. Protrusion and Laterotrusion According to Sex, Age, Height, Weight, BMI, and MW

Valid percentage measurements for P, RL, and LL were relatively low due to a lack of compliance during measurements, especially in the young children (Table A1). While males showed higher LL values (*p* < 0.05), P and RL values did not significantly differ between males and females (*p* > 0.05, Table 1). ANOVA followed by an LSD post hoc test showed that differences in the quartile groups in terms of P, RL, and LL were not statistically significant (*p* > 0.05, Table A2).

### 3.4. Correlation Analysis

Hinkle et al.’s classification [9] of the correlation coefficient with practical magnitude was used in this study; a moderate positive correlation (correlation coefficient (r), 0.50–0.70) was observed between the MMO and age, height, weight, and MW, with the highest correlation between MMO and height (*p* < 0.001). In contrast, the MA showed a moderate positive correlation with MW, and low positive correlations (r, 0.30–0.50) with age, height, weight, and BMI (*p* < 0.001). However, there was no significant correlation between P, RL, and LL and age, height, weight, BMI, and MW. In addition, a moderate positive correlation was found between the overjet with P (r = 0.497; *p* < 0.001) (Table 2).

### 3.5. Multiple Linear Regression Analysis

Age and height had high variance inflation factor (VIF) values (12.363 and 17.257, respectively) (Table A3), as observed in the very high positive correlations between these two variables (correlation coefficient, 0.953; *p* < 0.001). Therefore, height and weight were substituted for BMI. The final regression model with four variables, namely, sex, age, BMI, and MW, then had small VIF values of between 1.0 and 3.0, implying that there was no issue of multicollinearity. The fit of this final model was then adjusted (the adjusted R^2^ slightly decreased from 0.441 to 0.432, and the F statistics increased from 69.204 to 83.280). In the multivariate analysis, the MMO increased with age, BMI, and MW (*p* < 0.001) (Table 3). 

Multiple linear regression analysis of the MA showed that sex and MW had a significant influence on the MA (R^2^ = 0.367, adjusted R^2^ = 0.360, F = 59.034, *p* < 0.001). In other regression analyses involving P, RL, and LL as dependent variables, the adjusted R^2^ values were not remarkably high (0.018, 0.023, and 0.022, respectively).

## 4. Discussion

Limited mouth opening is an important sign in the diagnosis of TMD [7,10], and certain diseases involving the TMJ, such as juvenile idiopathic arthritis, which are especially common in children and frequently present with restricted mouth opening [11,12]. Moreover, since children possess a limited ability to describe discomfort and pain, localize their painful symptoms, or understand questions related to their pain, true TMDs may be overlooked. Thus, clinical examinations that are simple and quick to perform can be practically significant. Hence, TMJ ROM measurements can be conveniently used in TMD screening examinations for baseline function evaluation. 

Maximal opening of the mouth in the supine position can narrow the upper airway diameter [2], which contributes to upper-airway constriction, especially during midazolam sedation [13]. Furthermore, in terms of anatomy, children have relatively large tongues and short necks compared with adults [14]; therefore, mouth props in pediatric dental procedures should be used carefully, warranting safe guidelines for applying mouth props in dental practice.

Korean children and adolescents encompassing the entire age range were recruited for unassisted MMO measurement; the distribution of participants was nearly equal (*n* = 15, according to sex; *n* = 30, according to age). The presence of TMD was investigated and participants with any pain or restriction in terms of mandibular movement were excluded in this study [15]. However, participants with TMJ sounds or bruxism habits without any symptoms of TMDs were included because joint sounds are common in the general population [16] and not all sounds need to be treated [17]. To avoid bias due to incomplete eruption of the central incisors, only fully erupted central incisors were included. Following DC/TMD, all measurements were recorded upright in a chair to ensure the participants’ comfort and compliance. Visscher et al. reported that head posture could influence the intra-articular distance in the TMJ [18]; therefore, ROM measurements were performed in an upright position with the head positioned in a natural head posture and supported using the dental chair headrest. This may have contributed to the high level of internal consistency for the MMO, P, RL, LL, and MA values. 

Although vertical movements in DC/TMD were corrected on addition of the vertical overlap [7], positive overbite during unassisted MMO was not considered in this study to record the measurements quickly and conveniently to ensure and enhance the participants’ compliance and attention. To minimize errors due to overbite, participants with negative overbite or crossbite were excluded. Horizontal overlap was considered as suggested in the DC/TMD.

Statistically significant differences between the sexes were observed in terms of MMO, MA, and MW, which differed from the results reported in previous studies [6,19,20]. This may have been related to the statistically significant difference in height and weight between males and females (*p* < 0.01 and *p* < 0.001, respectively). These differences were attributed to sexual maturity occurring earlier in adolescents, since height and weight were significantly different only among the 14- and 15-year-old adolescent participants in this study compared to those in previous studies.

The MMO measurements increased consistently with age, which has also been reported in previous studies [3,4,6]. This can be partially explained by the mandibular growth increasing with age, geometrically affecting the MMO [1,4,19]. This also supports the correlation results in that the MMO had a higher correlation with height (r = 0.633) than with chronological age (r = 0.609). These correlation coefficients were relatively high, showing a moderate positive correlation compared with the correlation coefficients reported in prior studies (r = 0.4) [6], which was attributed to the even age and sex distribution in this study. MA showed similar tendency with MMO. In clinical practice, planar MA rather than linear MMO is more meaningful for an accurate oral examination, especially during orthodontic therapy, which can cause gingival and mucosal lesions [21]. A thorough search of the relevant literature yielded no article to measure MA in addition to vertical and horizontal movements.

In this study, the cut-off point of MMO where 40 mm is reached in a young population was obtained. A restricted mouth opening has been traditionally assessed using less than 40 mm as a reference. In the DC/TMD, disc displacement without reduction with or without limited opening is determined by maximum assisted opening of <40 mm [7], the minimum for normal mouth opening [22]. Only 1.2% of the adults showed restricted opening [23], since healthy adults open at a maximum of 40 mm and more. Therefore, the cutoff where 40 mm was reached by age, height, and weight was investigated. These values were similar to those reported in previous studies that showed even 6-year-old children are able to open their mouth to ≥40 mm [20,23]. On applying these findings to pediatric dental procedures, one should be more cautious during forcible mouth opening over 40 mm, especially in children shorter than 105 cm, lighter than 18 kg, and under 5 years old. 

However, a single cut-off value does not appear to be adequate for the definition of limited ROM, especially in a growing population, considering inter-individual differences in terms of sex, age, height, weight, and facial morphology. In each age group, a wide range of MMO (ranging from 25 mm to 57 mm in 3-year-old children, and from 32 mm to 63 mm in 15-year-old adolescents) was found, which was similar to results reported in previous studies [4,5]. Therefore, it is confirmed that the standard reference should be expressed as a standard range of normalcy with consideration to personal characteristics. Moreover, despite a healthy adult’s MMO being reached at an early age, MW and MA in children remain significantly smaller than those of adults, and this should be considered when using dental instruments.

Even children aged 3 to 6 years could normally move 6 mm and more on protrusive and lateral excursions when they are able to understand and perform mandibular movements as instructed. However, there were limitations in obtaining accurate anterior and lateral movements for children aged <7 years owing to the challenges encountered in ensuring that children follow directions and due to a lack of cooperation, which has also been reported in previous studies [19,20]. Therefore, several measurements were lacking, and P, RL, and LL did not show a close relationship to sex, age, height, weight, and MW.

This study had several limitations. First, overbite during MMO measurement was not considered. Second, all the measurements in this study were recorded with participants in a seated position. More information is needed to correlate ROM measurements in sitting and supine position. ROM and MA in supine, upper airway dimensions, and air flow should be studied in future research projects. In addition, further studies of MA in adults will provide more data to utilize MA in the young population in clinical practice.

## 5. Conclusions

This study investigated TMJ ROM, and MA values obtained from children and adolescents aged 3–15 years old of homogenous ethnicity, and found high correlation with age, height, weight, BMI, and MW. The findings of this study suggest that forcible mouth opening over 40 mm should be cautiously considered especially in children shorter than 105 cm, lighter than 18 kg, and under 5 years. Further investigations are required to correlate ROM and MA in sitting to evaluate the perils of mouth props.

## Figures and Tables

**Figure 1 children-08-00515-f001:**
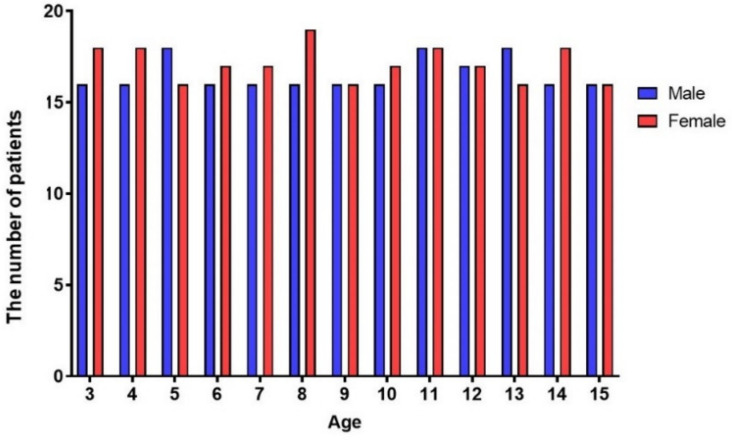
Distribution of participants according to sex and age.

**Figure 2 children-08-00515-f002:**
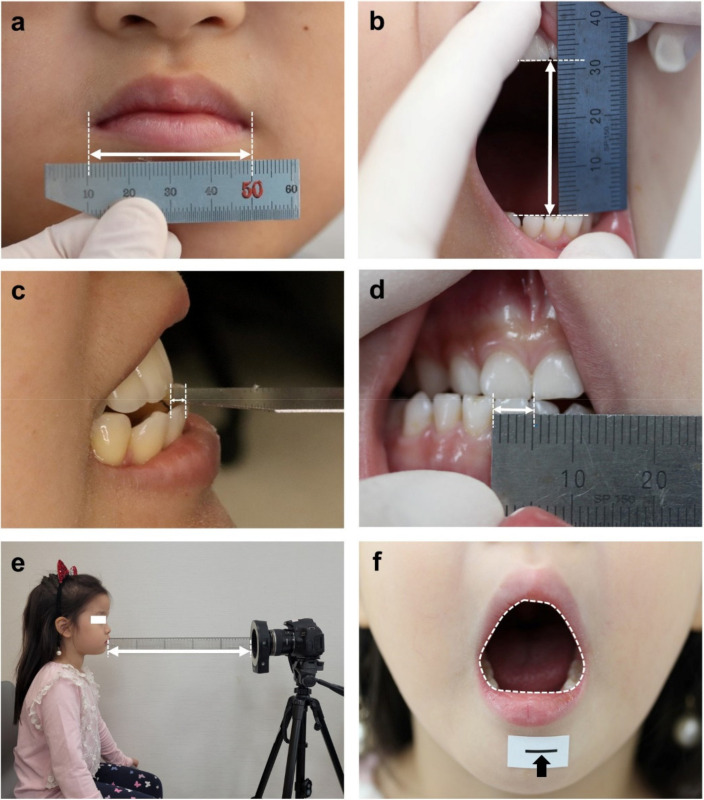
Measurement of mouth width (**a**), maximum mouth opening (**b**), protrusion (**c**), right laterotrusion (**d**), photo room setting (**e**), and mouth area (**f**). Mouth area was measured at the same location with a background indicating a distance of 50 cm. The measurement of mouth area was recorded in the inner area when the outline of the inner lip line was connected, and calculated by placing a sticker that sets the standard of 10 mm (black arrow).

**Figure 3 children-08-00515-f003:**
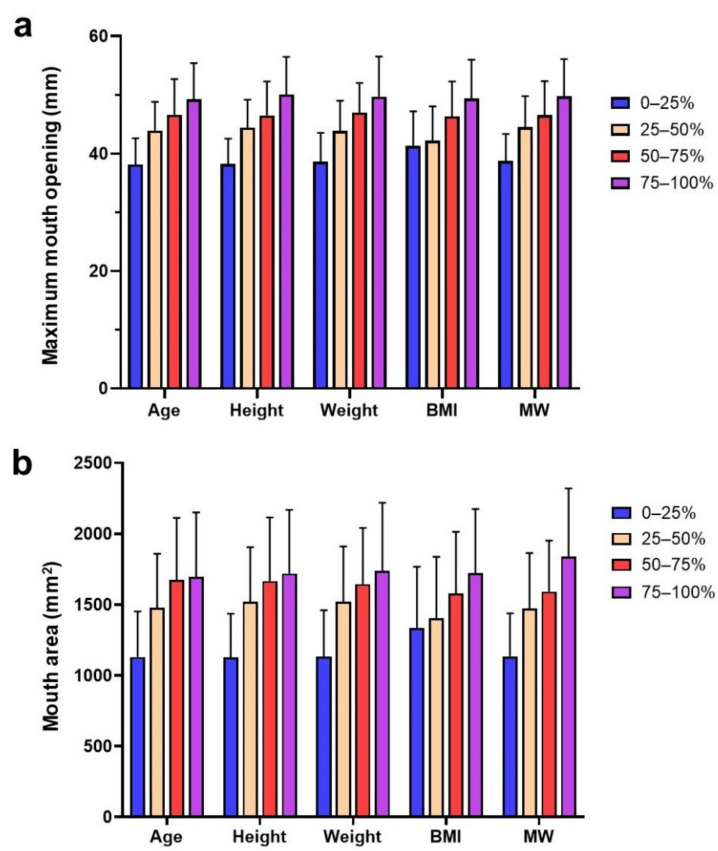
Range distributions of maximum mouth opening (**a**) and mouth area (**b**) in quartile groups according to the studied age, height, weight, body mass index and mouth width. BMI, body mass index; MW, mouth width.

**Figure 4 children-08-00515-f004:**
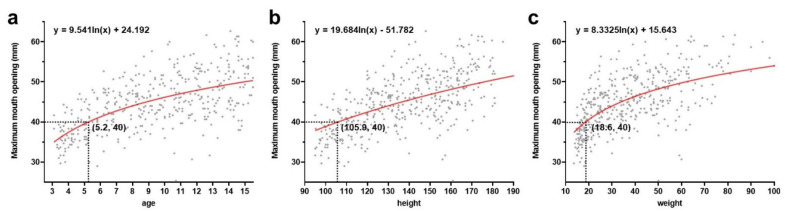
Logarithmic trend line of maximum mouth opening according to the studied age (**a**), height (**b**), and weight (**c**).

**Table 1 children-08-00515-t001:** Baseline participant data using a two sample *t*-test.

Variables	Total (*n* = 438)	Male (*n* = 215)	Female (*n* = 223)	*p*
Mouth width (*n* = 433)	43.1 ± 5.4	43.9 ± 5.6	42.3 ± 5.0	0.003 *
Maximum mouth opening (*n* = 437)	44.8 ± 6.9	45.9 ± 7.6	43.8 ± 6.0	0.001 *
Protrusion (*n* = 344)	7.1 ± 2.1	7.1 ± 2.2	7.0 ± 2.1	0.583
Right laterotrusion (*n* = 330)	7.6 ± 2.1	7.8 ± 2.0	7.3 ± 2.2	0.060
Left laterotrusion (*n* = 333)	7.5 ± 2.3	7.8 ± 2.3	7.2 ± 2.1	0.015 *
Mouth area (*n* = 417)	1511.6 ± 463.2	1613.5 ± 513.9	1412.0 ± 383.5	<0.001 *

Mouth width, maximum mouth opening, protrusion, right and left laterotrusion values are in millimeters, and mouth area values are in square millimeters. Values are presented as mean ± standard deviation. Two sample *t*-test. * *p* < 0.05.

**Table 2 children-08-00515-t002:** Correlation coefficients between mandibular range of motion and mouth area with age, height, weight, body mass index, mouth width and overjet.

	MMO	P	RL	LL	MA
**Age**	0.609 **	0.162 **	0.095	−0.010	0.459 **
**Height**	0.633 **	0.136 *	0.077	−0.026	0.477 **
**Weight**	0.621 **	0.121 *	0.085	−0.011	0.451 **
**BMI**	0.481 **	0.127 *	0.084	0.047	0.353 **
**Mouth width**	0.604 **	0.141 **	0.169 **	0.101	0.585 **
**Overjet**	0.121 *	0.497 **	0.168 **	0.169 **	0.174 **

** Pearson’s *p* < 0.01. * Pearson’s *p* < 0.05. MMO, maximum mouth opening; P, protrusion; RL and LL, right and left laterotrusion; MA, mouth area.

**Table 3 children-08-00515-t003:** Multiple linear regression analysis of the association between maximum mouth opening and sex, age, BMI, and MW.

	Unstandardized Coefficients		Std. Coefficients			Collinearity Statistics	
	β	Std. Error	β	t	*p*	Tolerance	VIF
(Constant)	18.732	2.804		6.680	<0.001 *		
Sex	−1.011	0.518	−0.074	−1.951	0.052	0.911	1.098
Age	0.504	0.115	0.275	4.406	<0.001 *	0.337	2.967
BMI	0.285	0.082	0.158	3.491	0.001 *	0.392	2.550
0Mouth width	0.391	0.073	0.308	5.328	<0.001 *	0.643	1.555

R^2^ = 0.437, Adjusted R^2^ = 0.432, F = 83.280, <.001; * *p* < 0.05; Std, standardized; VIF, variance inflation factor.

## Data Availability

The data presented in this study are available in insert article or Appendix A.

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
