# Peer review of "An Analysis of the Temporomandibular Joint Range of Motion and Related Factors in Children and Adolescents"

_children, 2021, doi:10.3390/children8060515_

Round 1

Reviewer 1 Report

The article is well written and with a correct scientific methodology. It is of great interest to allow us to formulate guidelines in the approach to TMJ in pediatric age. I suggest the authors to read the following articles: "M. Roncati, et al. An oral health aid for disabled patients. Dental Cadmos. 2013; 81 (7), 447-452."; "Lucchese A, et al. Everstick® and Ribbond® fiber reinforced composites: Scanning Electron Microscope (SEM) comparative analysis. European Journal of Inflammation. 2011; 9, 73-79." and "Oral mucosal complications in orthodontic treatment. Manuelli M, et al. Minerva Stomatol. 2019 Apr; 68 (2): 84-88.", which constitute guidelines in further problems of dental interest in pediatric age and during orthodontic treatment. 

Author Response

I would like to express my appreciation for your advice.

We agreed with your comments and introduced corrections where required in the manuscript.

I look forward to your response and hope that the revised manuscript is suitable for publication in your journal.

Yours sincerely,

Chung-Min Kang

Reviewer 2 Report

The paper itself is well written showing a great effort from the authors.

The topic does not sound so original and maybe instrumental equipment such as a sliding caliper or even better kinesiography could be more accurate in assessing these measurements compared to a ruler.

I also would make only the following few mentions:

Line 68: How did you judge "an abnormal facial profile"?

Line 70: What does it mean "precocious puberty" and why it was excluded?

FIG.2 B: It could be better to use a figure where the operator wears gloves while taking measurements.

Line 105: It could be useful to add a figure illustrating how laterotrusion was calculated. This is the only measurement that was not explained by a figure.

Line 110: Which camera did you used to take photographs? Please add in matherial and methods section.

Line 111: How did you measure the 50 cm? It does not seem so reproducible.

Author Response

(The authors gave the same response as above.)

Reviewer 3 Report

I had the opportunity of revising the present manuscript regarding An Analysis of the Temporomandibular Joint Range of Motion and Related Factors in Children and Adolescents.

The study is interesting and could merit publication after a proper revision on some points.

I list here my observations:

  1. Please avoid personal pronouns in the whole manuscript and use impersonal forms.
  2. The introduction section should be reorganized to give a rationale to the study and guide the reader onto the aim of the study. The sentence "In this study, we tried to 38 obtain the cut-off point of MMO where 40 mm is reached in a young population."should be moved to the discussion as the sentence "In this study, MA was intro-42 duced and defined as the area of the mouth entrance when MMO was achieved. "
  3. I suggest to change "investigated" with "measured" in page 3 line 101
  4. Was the presence of TMD investigated in the sample? A previous study showed that in adolescents the presence of TMD (or TMJ alterations limited in time duration) is quite frequent. I suggest to discuss this point as it could influence the results of the study if the provided ROM data would be considered as normative values. Tecco S, Nota A, Caruso S, Primozic J, Marzo G, Baldini A, Gherlone EF. Temporomandibular clinical exploration in Italian adolescents. Cranio. 2019 Mar;37(2):77-84. doi: 10.1080/08869634.2017.1391963.

Best Regards

Author Response

(The authors gave the same response as above.)

Round 2

Reviewer 3 Report

The authors improved the manuscript performing most of the suggested changes. Anyway many personal forms are still present.

Please change personal forms to impersonal (i.e. personal pronouns as "we" or "our" should be replaced by impersonal forms)

Best Regards

Author Response

(The authors gave the same response as above.)
